# Using Femtosecond Laser Pulses to Explore the Nonlinear Optical Properties of Ag/Au Alloy Nanoparticles Synthesized by Pulsed Laser Ablation in a Liquid

**DOI:** 10.3390/nano14151290

**Published:** 2024-07-31

**Authors:** Yasmin Abd El-Salam, Hussein Dhahi Adday, Fatma Abdel Samad, Hamza Qayyum, Tarek Mohamed

**Affiliations:** 1Laser Institute for Research and Applications LIRA, Beni-Suef University, Beni-Suef 62511, Egypt; 2Laser-Matter Interaction Laboratory, Department of Physics, COMSATS University Islamabad, Park Road, Islamabad 45550, Pakistan; 3Department of Engineering, Faculty of Advanced Technology and Multidiscipline, Universitas Airlangga, Surabaya 60115, Indonesia

**Keywords:** laser ablation, metal alloy NPs, metal NPs, surface plasmon resonance, femtosecond laser, Ag–Au alloy NPs

## Abstract

Metallic nanoparticles have gained attention in technological fields, particularly photonics. The creation of silver/gold (Ag/Au) alloy NPs upon laser exposure of an assembly of these NPs was described. First, using the Nd: YAG pulsed laser ablation’s second harmonic at the same average power and exposure time, Ag and Au NPs in distilled water were created individually. Next, the assembly of Ag and Au NP colloids was exposed again to the pulsed laser, and the effects were examined at different average powers and exposure times. Furthermore, Ag/Au alloy nanoparticles were synthesized with by raising the average power and exposure time. The absorption spectrum, average size, and shape of alloy NPs were obtained by using an ultraviolet-visible (UV–Vis) spectrophotometer and transmission electron microscope instrument. Ag/Au alloy NPs have been obtained in the limit of quantum dots (<10 nm). The optical band gap energies of the Ag/Au alloy colloidal solutions were assessed for different Ag/Au alloy NP concentrations and NP sizes as a function of the exposure time and average power. The experimental data showed a trend toward an increasing bandgap with decreasing nanoparticle size. The nonlinear optical characteristics of Ag/Au NPs were evaluated and measured by the Z-scan technique using high repetition rate (80 MHz), femtosecond (100 fs), and near-infrared (NIR) (750–850 nm) laser pulses. In open aperture (OA) Z-scan measurements, Ag, Au, and Ag/AuNPs present reverse saturation absorption (RSA) behavior, indicating a positive nonlinear absorption (NLA) coefficient. In the close-aperture (CA) measurements, the nonlinear refractive (NLR) indices (*n*_2_) of the Ag, Au, and Ag/Au NP samples were ascribed to the self-defocusing effect, indicating an effective negative nonlinearity for the nanoparticles. The NLA and NLR characteristics of the Ag/Au NPs colloids were found to be influenced by the incident power and excitation wavelength. The optical limiting (OL) effects of the Ag/Au alloy solution at various excitation wavelengths were studied. The OL effect of alloy NPs is greater than that of monometallic NPs. The Ag/Au bimetallic nanoparticles were found to be more suitable for optical-limiting applications.

## 1. Introduction

Nonlinear optical (NLO) materials have attracted substantial interest from researchers, especially with the progress of ultrashort laser pulses [1]. These materials hold pivotal significance in the advancement of all-optical and electro-optical systems, along with various applications in optical communications, light-induced chemical reactions, and optical computing [2,3,4,5]. Nanostructured materials stand out among NLO materials due to their distinct and highly sought-after attributes, such as large surface-to-volume ratios and distinct structures, which often yield superior electrical and optical attributes than their bulk counterparts. Metal nanoparticles (NPs) have emerged as a focal point of research due to their exceptional optical properties and their diverse applications across various fields, including biosensing, drug delivery, catalysis, and physiochemical and optoelectronic domains, as well as surface-enhanced Raman scattering and detection [6,7,8,9,10,11,12,13]. Quantum dots (QDs) are made of very small metal particles, a thousand times less than the size of a hair. Metal and alloy nanoparticles in the quantum size regime have attracted attention in recent years due to their ability to size-dependently control electrical, optical, and catalytic capabilities, as well as the potential for technological advancement [14,15,16,17,18]. Offering a versatile range of emitted light from ultraviolet to infrared, typically within the diameter range of 2 to 10 nm [19]. In recent times, alloy NPs have attracted more interest owing to their unique characteristics and wide range of applications [20,21,22,23]. Notably, silver (Ag)/gold (Au) alloy nanostructures have found extensive utilization across diverse fields, including antibacterial applications [24], bioimaging [25], biolabeling [26], catalysis [27,28], drug delivery, cancer therapy [29], nanophononics [30], optoelectronics [31], and nanoscale optical biosensors [32].

Numerous methods, classified as top-down and bottom-up techniques, are utilized to create Au and Ag NPs. These techniques include chemical, physical, and biological approaches [33,34]. Among these, a common technique involves the coproduction of metal precursor salts; however, this method often falls short in yielding purified NPs, necessitating chemical purification and the use of potentially hazardous reagents [34].To overcome these constraints, a novel, adaptable, and very effective physical technique known as pulsed laser ablation in liquid (PLAL) of a bulk target has gained favor as an approach to creating NPs, offering a wide array of optical properties, compositions, and morphologies [35,36]. This physical method, which usually follows a top-down approach, involves separating small particles (atoms, ions, etc.) from bulk materials to produce small nanoclusters.

In laser-based methods, the bulk material is heated by a laser, leading to its evaporation followed by vapor condensation, ultimately forming nanoparticles [37]. The efficacy and characteristics of the ablation process, including particle size distribution and structure, are influenced by factors such as the medium, experimental geometry, and laser parameters (pulse duration, wavelength, repetition rate, and energy) [38,39,40,41]. Many studies have explored the creation of alloy NPs via laser irradiation combined with two different metal nanoparticles. This technique is known as pulsed laser irradiation in liquid (PLIL) [42,43,44,45]. Izgaliev et al. described the synthesis of colloidal solutions of Au and Ag NPs separately through target ablation in a liquid medium, followed by irradiation of the combined Ag and Au NPs with a pulsed laser to produce Au–Ag alloy NPs [42,46]. Similarly, Hajismailbaiji et al. achieved the generation of Au–Ag alloy NPs by irradiating a combination of independently produced monometallic colloidal suspensions [46]. Qayyum et al. investigated the influence of 1064 nm and 532 nm lasers on Ag/Au alloy NP formation, observing a direct correlation between the alloying process and the laser wavelength utilized [37]. Lee et al. discussed the creation of Ag/Au alloy NPs employing pulsed laser ablation of bulk metals used in alloys within water, marking significant advancements in these burgeoning techniques [47].

Vincenzo et al. presented how the plasmonic properties of bilayers of Ag on Au nanoporous films can lead to an effective dielectric permittivity much larger than that of the original constituent metals, allowing the creation of nanoporous metal layers with tailored optical responses [48]. 

Despite progress, a comprehensive understanding of the nonlinear optical (NLO) attribute properties of Au–Ag nanoparticles, particularly the effects on these nanostructures (mono- and bimetallic) when their morphology and composition vary, remains elusive [49,50,51,52,53]. This gap persists due to the significant changes in their properties under such alterations, necessitating further investigation.

In this paper, we report the synthesis of Ag/Au alloy NPs via post-laser reirradiation of a mixture of independently generated Ag and Au NPs using the PLAL method. We investigated the effect of laser irradiation parameters such as laser intensity and exposure time on the alloying process. The Ag/Au alloy NPs were characterized using UV–Vis spectrophotometry, transmission electron microscopy, and inductively coupled plasma analysis.

Furthermore, both open-aperture (OA) and closed-aperture (CA) Z-scan techniques with femtosecond laser pulses were used to experimentally investigate the nonlinear refractive index (*n*_2_) and nonlinear absorption coefficient (γ) of the Ag NPs, Au NPs, and Ag/Au alloy NPs. Nonlinear optical (NLO) properties of the colloidal Ag/Au alloy have been studied at different excitation wavelengths between 750 and 850 nm and at different incident laser powers between 0.6 and 1.2 W. Additionally, we investigated the optical limiting (OL) consequences of solutions that contained Ag NPs, AuNPs, and Ag/Au alloy NPs. Particular attention was paid to studying how the excitation wavelength affected the OL behavior of the Ag/Au alloy NPs.

## 2. Experimental Setup

### 2.1. Laser Ablation Setup

Figure 1 displays the layout of the experimental equipment for creating NPs using PLAL. Ablation was performed using an Nd: YAG laser (Quanta-Ray Pro-350) with a wavelength of 532 nm, a pulse duration of 10 ns, and a repetition rate of 10 Hz. To obtain Ag/Au alloy NPs by PLAL, square pieces of target Ag and Au with a high purity of approximately 99.99% and dimensions of 20 × 20 × 2 mm were exposed individually to the pulsed laser ablation process. Subsequently, Ag/Au alloy NPs were produced by reirradiation of a combination of NPs from a colloidal Ag and Au solution using a pulsed laser.

The initial step in the production of Ag/Au alloy NPs was to use PLAL to create Ag and Au NPs. There were a few procedures that had been followed. Before the ablation process, the target Ag and Au were polished to remove the oxide layer caused by air exposure. The bigger surfaces of Ag and Au should be smooth and burr-free. The Ag and Au were then ultrasonically cleaned for 30 min with ethanol and deionized water to eliminate any organic residues. Next, 10 mL of distilled water was added to a beaker, and the Ag or Au target at the bottom was immersed. The beaker was mounted to the monitoring speed device. To prevent the laser beam from centering on the same spot in the sample or from etching the sample, this device simultaneously spins the beaker and the target. Furthermore, the speed was set at 177 RPM, and the target was positioned at a distance about equivalent to the focusing distance of a 10.5 cm convex lens with a laser beam diameter of 0.35 cm.

The Ag and Au NPs were prepared by adjusting the exposure time and intensity to 30 min and 71.9 MW/cm^2^, respectively. The Ag/Au alloy nanoparticles were synthesized utilizing the same setup as shown in Figure 1, without the convex lens. At this point, a 1:1 mixture of 6 mL of Ag NPs colloid and 6 mL of AuNPs was used. Following that, the mixture was exposed to another round of pulsed laser radiation for varying exposure durations (5–30 min) and intensities of 6.3, 8.9, and 11.4 MW/cm^2^.

The characterization of the formed Ag, Au, and Ag/Au alloy NPs was performed using an ultraviolet-visible (UV–Vis) spectrophotometer (Model: C-7200) to record the absorption spectra, and the concentration of colloidal NPs was measured by an inductively coupled plasma (ICP) (Agilent 5100 Synchronous Vertical Dual View (SVDV) ICP-OES, Agilent Vapor Generation Accessory VGA 77). Furthermore, the shape and average size distribution of the Ag/Au alloy NPs were identified using transmission electron microscopy (HR-TEM, JEM-2100, Joel, Japan, operated at 200 KV).

### 2.2. Z-Scan Setup

The Z-scan setup employed to investigate the nonlinear optical characteristics of Ag, Au, and Ag/Au metal alloy NPs is exhibited in Figure 2. In the present study, the experimental Z-scan setup used the laser pulses system (INSPIRE HF100) from Spectra-Physics, which was operated by a femtosecond (fs) Ti: sapphire laser (MAI TAI HP) from Spectra-Physics with a 1.5 W–2.9 W average power, an 80 MHz repetition rate, and a 690 nm–1040 nm range of wavelength [13]. The INSPIRE HF100 operates at IR Ti: sapphire pump wavelengths and has two additional modes of operation based on NLO phenomena: (i) a second harmonic generator (SHG) and (ii) an optical parametric oscillator (OPO) [54,55]. These modes of operation produce wavelengths ranging from 345 nm to 2500 nm, which are covered by four exit apertures. The colloidal solution NPs were exposed to 100 fs Gaussian laser pulses at various wavelengths ranging from 750 nm to 850 nm, selected by the aperture of the fundamental IR pump. The INSPIRE laser beam has a profile of a Gaussian distribution with a TEM00 spatial mode and M2 < 1.1. A 5-centimeter convex lens is utilized to tightly focus the laser pulses. Metal NP colloids were placed in a quartz cuvette with a path length of 1 mm, which is less than the Rayleigh length (z_0_). The sample was scanned around the focus using a micrometer translation stage. The transmitted intensity of the colloidal solutions was measured by a power meter (PM1, Newport 843 R) as a function of sample position relative to the focus. In the OA setup, the normalized transmittance is measured by adjusting the aperture size (S) fully open (S = 1). The change in OA Z-scan measurement is related to the nonlinear absorption coefficient γ. In the CA setup, the laser beam that passes through the sample is directed through a closed aperture just before the power meter (PM2, Newport 843 R). This aperture is designed to ensure that any nonlinear phase shift in the beam is caused only by the colloidal solution. The present experimental measurements have a 10% uncertainty, which is mostly due to the determination of the irradiance distribution employed in the experiment, namely laser power calibration, pulse width, and beam waist.

## 3. Results and Discussion

### 3.1. Synthesis of Ag and Au Nanoparticles

Figure 3 shows the UV-vis absorption spectra measured via spectrophotometry of Ag and Au NPs synthesized using PLAL at a laser intensity of 71.9 MW/cm^2^ and an exposure time of 30 min. The distinct colors observed in the Ag and Au NP colloids are indicative of their composition and microstructure [56], with each solution exhibiting a characteristic fingerprint color. According to Mie’s theory [57], surface plasmons—a phenomenon where oscillating electromagnetic radiation of incident light causes collective oscillations of free electrons—are responsible for this coloration. The surface plasmon resonance (SPR) of Ag and Au NPs were observed at 409 and 506 nm, respectively, and are shown in Figure 3. The spherical shape of the NPs is indicated by the appearance of a single SPR peak [58,59].

Ag NPs display stronger actual absorbance with higher plasmon energy than those of AuNPs, as reported in the literature [24].

The Ag and Au NPs concentrations were quantified at 15 mg/L ± 1.5 mg/L and 19.2 mg/L ± 1.9 mg/L, respectively, using an inductively coupled plasma (ICP). Furthermore, we used transmission electron microscopy (TEM) to identify the NPs’ shape and characterize their size distribution. At room temperature, the colloidal samples were spread out on copper grids coated with carbon and left to air dry. Particle sizes of both Ag and Au NPs were determined using ImageJ software, with diameters measured from multiple distributed particles in TEM images [60]. Additionally, the software was employed to generate histograms and calculate the average nanoparticle size. Figure 4a,b depict histograms showing the Ag and Au NPs size distribution, respectively, with TEM images inset, confirming the spherical shape of the NPs. The Ag and Au NPs were found to have average sizes of 17.9 nm ± 1.8 nm and 5.1 nm ± 0.5 nm, respectively. The TEM images show that Au NPs have an average size of less than 10 nm, which is within the range of quantum dots.

Remarkably, Ag and Au NPs show slightly distinct average and size distributions despite being produced under the same conditions. This variation may be attributed to inherent physical characteristics of the materials, such as differences in melting points, absorption of laser light, thermal conductivity, and boiling points.

### 3.2. Synthesis of Ag/Au Alloy NPs

#### 3.2.1. Linear Optical Properties

After completing the first step, which produced Ag and Au NPs, attention turned to the creation of Ag/Au alloy NPs, or the second stage. This stage involved studying the interactions between Ag and Au NPs using a 1:1 volume ratio of colloidal Ag and Au NPs. Furthermore, the combination’s absorption spectra were recorded before reirradiation with a pulsed laser. Figure 5 depicts the absorption spectra of colloidal Au, Ag, and Ag/Au alloy NPs in the wavelength range of 200 nm–1100 nm. Notably, there are clear peaks in the spectra that represent the Ag NPs, Au NPs, and the combined colloids Ag/Au NPs. These peaks coincide with the Ag and Au NP SPR absorption peaks, confirming the bimodal nature of the spectrum. Figure 5 displays a decrease in the absorption spectrum of the combined colloids (Ag/Au NPs), attributable to the differential volumes of NPs utilized in the UV-Vis absorption spectrophotometer. For each measurement, each Ag and Au NPs UV–Vis absorption spectrum was measured using a 2 mL colloidal solution. Thus, 1 mL of Ag NPs solution and 1 mL of Au NPs solution were added to the colloid mixture to maintain a constant total volume. As a result, half the volume of each type of NPs colloidal solution reduced the overall absorption.

After combining Ag and Au NPs with a volume ratio of 1:1, experiments were undertaken to explore the effects of changing laser intensity and exposure time on the formation of Ag/Au NPs; however, this is not an alloy yet.

The effect of reirradiating a combination of Ag/Au NPs at various laser intensities, 6.3, 8.9, and 11.4 MW/cm^2^, for a constant exposure time of 15 min was initially investigated. Figure 6a illustrates the Ag/Au NPs alloy’s UV–Vis absorption spectrum with various laser intensities.

At a laser intensity of 6.3 MW/cm^2^, separate peaks were observed representing Ag and Au NPs with a bandwidth (Δλ) of 0.18 μm ± 0.02 μm. Additionally, a slight redshift was noted in the SPR for the Ag NPs, while a blueshift was observed for the Au NPs. This shift showed that the optical qualities of the colloidal solution NPs were changed, illustrating how differences in the optical or morphological properties of the colloidal solution NPs may affect the UV absorption spectrum [37].

At a laser intensity of 8.9 MW/cm^2^, the two plasmon peaks of the Ag and Au NPs were found to shift, and the bandwidth dropped to 0.14 μm ± 0.01 μm, which was similar to that at 6.3 MW/cm^2^. Furthermore, as the laser intensity of reirradiation of the Ag/Au NPs combination increased to 11.4 MW/cm^2^, the bandwidth reduced further to 0.13 μm ± 0.01 μm from values shown at 6.3 to 8.9 MW/cm^2^. Remarkably, at 441 nm, a quite narrow single peak appeared. This gradual transition from a bimodal colloidal peak to a narrow single SPR absorption peak validated the formation of Ag/Au alloy NPs. The Ag/Au alloy was identified by a single SPR peak, which is consistent with the results of other studies [37,42,60,61].

In the second phase, as shown in Figure 6b, we studied the impact of different exposure times (from 5 to 30 min) on the reirradiation of Ag/Au alloy NPs at a constant laser intensity of 8.9 MW/cm^2^. At 5 min, there was a broader absorption peak with a Δλ of 0.22 μm ± 0.02 μm, accompanied by two distinct SPR peaks attributed to Ag and Au NPs. With an increase in irradiation time to 15 min, the bandwidth of the peak decreased to a value comparable to that at 5 min, reaching 0.14 μm ± 0.01 μm. Furthermore, the presence of two peaks related to Ag and Au NPs was noted. Subsequently, when the reirradiation time was extended to 30 min, there was a broadening of the bandwidth to 0.13 μm ± 0.01 μm, accompanied by a decrease in peak intensity to a level comparable to that at 15 min. Additionally, a relatively single peak emerged at 454 nm. This progressive transition from a bimodal colloidal peak to a single SPR absorption peak confirmed the formation of Ag/Au alloy NPs [37,42,60,61].

#### 3.2.2. Characterization of the Average Size and Structure

The average size and structure of the Ag and Au NPs colloidal mixture were assessed using transmission electron microscopy (TEM). Figure 7 illustrates the size distribution histogram generated from laser-induced irradiation of a combination of colloidal Ag and Au NPs at various laser intensities while maintaining a constant exposure time of 15 min. Additionally, the TEM image is presented in the insets of Figure 7a–c, demonstrating the spherical structure of the mixture.

Through the reirradiation of a mixture of colloidal Ag and Au NPs, the effects of varied laser intensities at constant exposure times and different exposure times at a constant laser intensity were investigated and shown in Figure 7 and Figure 8, respectively. As shown in Figure 7, after reirradiation of the mixture with laser pulses at laser intensities of 6.3, 8.9, and 11.4 MW/cm^2^, the average sizes observed were 10.1 ± 1, 8.9 ± 0.8, and 8.5 ± 0.8 nm, respectively. During this stage, Ag/Au alloy NPs were formed with a laser intensity of 11.4 MW/cm^2^, as previously observed. As shown from these results for the average sizes of Ag, Au, and Ag–Au NPs 17.9, 5.1, and 8.5 nm, respectively, average sizes of Ag–Au alloy are typically produced among their metals. Qayyum et al. [37] determined the average diameter of Ag and Au NPs is 25 nm and 18 nm, and Ag–Au alloy NPs with a mean diameter of 24 nm. Mamta et al. [62] reported average sizes of NPs obtained to be 15 nm, 3 nm, and 6 nm for AgNPs, CuNPs, and Ag@Cualloy NPs, respectively.

Figure 8 shows the size distribution histogram and the structure of the Ag/Au alloy NPs. Figure 8a–c depict changes in the average sizes at 11.1 ± 1, 8.9 ± 0.8, and 8 ± 0.8 nm, respectively, with varying exposure times of 5, 15, and 30 min at the same laser intensity of 8.9 MW/cm^2^. As evident from these figures, the average size reduced as the exposure time increased to 30 min. This is consistent with previous observations, at an exposure time of 30 min, Ag/Au alloy NPs were formed during this stage.

### 3.3. Studying the Nonlinear Optical Properties of Ag/Au Alloy NPs

#### 3.3.1. Studying the Nonlinear Absorption Coefficient β of Ag/Au Alloy NPs

The nonlinear optical properties of Ag/Au alloy solutions, prepared via laser ablation and irradiated with laser intensity of 11.4 MW/cm^2^, a 532 nm incident wavelength, and 15 min of exposure time, were investigated. Figure 9 illustrates the experimental OA Z-scan measurements of Ag/Au alloy NPs in the colloidal state, conducted using femtosecond (fs) pulses and a high repetition rate (HRR) laser at 80 MHz.

The NLO properties of the Ag/Au colloidal solution were explored at different incident fs laser powers ranging from 0.6 to 1.2 W and excitation wavelengths spanning from 750 to 850 nm. All curves displayed in Figure 9a demonstrate reverse saturable absorption (RSA), which indicates an intensity-dependent absorption effect with the lowest transmission at the focus (valley). This symmetric transmission around the focus (Z = 0) signifies the distinctive signature of RSA, characterized by a decrease in transmittance with increasing input intensity. Various processes, including transient absorption, nonlinear absorption (NLA) due to interband transition, photoejection of electrons, and nonlinear scattering, are typically cited as influencing the effects of metallic NPs on RSA. The observed RSA behavior may stem from three-photon absorption (3PA), free carrier absorption (FCA), excited-state absorption (ESA), nonlinear scattering (NLS), or a combination of these processes. Given the bandgap of 3.49 eV for the Ag/Au alloy colloid, it is plausible to infer the occurrence of 3PA, thereby inducing a nonlinear absorption process.

To determine the nonlinear absorption coefficient, the OA Z-scan measurements were simulated using the NLO absorption model expressed by the following [63,64]:(1)$$T_{OA}=1\pm \left[\frac{\gamma \;{I_{0\;}}^{2}L_{eff}}{{\left(m+1\right)}^{\left(\frac{3}{2}\right)}{\left(1+\frac{Z^{2}}{{Z_{0}}^{2}}\right)}^{2}}\right]$$
where γ represents the 3PA coefficient, $I_{0}$ represents the peak intensity at the focus (Z=0), $Z_{0}$ is the Rayleigh length; $Z_{0}=\frac{\pi {\omega_{0}}^{2}}{\lambda }$, ($Z_{0}>L$), where λ is the excitation wavelength and *ω_o_* is the beam waist at the focus 19 μm ± 2 μm; and $L_{eff}=\left(1-e^{-m\alpha L}\right)/m\;\alpha$, where α is the linear absorption, m = 1 for two-photon nonlinear absorption (2PA) and m = 2 for 3PA, and L is the thickness of the sample. The normalized transmittance ($T_{OA}$) decreases when the incident power increases. Figure 9b depicts the effect of the laser incident power on the *γ* of the Ag/Au colloidal solution at an 800 nm excitation wavelength. As the incident power increases, γ decreases.

As the incident power rises, the population of free carriers also increases. This elevated concentration of electrons in the conduction band and holes in the valence band intensifies their interactions. Consequently, the frequency of collisions among free carriers escalates, leading to heightened scattering of photons and phonons, thereby causing a reduction in the NLA coefficient (γ).

NLS is most typically caused by the creation of two distinct types of scattering centers subsequent to the photoexcitation of NPs. Firstly, the excitation energy absorbed by the nanoparticle prompts rapid expansion, effectively creating the nanoparticle itself as a scattering center. Secondly, this absorbed energy is subsequently transferred to the surrounding solvent, inducing its heating and potentially forming bubbles, which act as secondary scattering centers.

Figure 10a depicts the variation of the nonlinear absorption coefficient with the excitation wavelength, ranging from 750 to 850 nm, for the Ag/Au alloy solution at a 1 W constant excitation power. The absorbance demonstrates an increasing trend as the excitation wavelength rises, indicating a concurrent increase in reverse saturable absorption (RSA). Furthermore, as the photon energy rises with increasing excitation wavelength, the likelihood of NLS augmentation increases, leading to a reduction in NLA [65]. The NLO properties of bimetallic NPs are notably influenced by their size and morphology. Figure 10b illustrates the nonlinear absorption coefficient plotted against the excitation wavelength. Remarkably, a linear increase in γ is observed with increasing excitation wavelength. This observation underscores the strong dependence of the NLA response on the excitation wavelength, suggesting a sensitivity of the NLA to both the excitation wavelength and incident intensity.

#### 3.3.2. Investigating the Nonlinear Refractive Index *n*_2_ of Ag/Au Alloy NPs

Figure 11a and Figure 12a display the experimental CA Z-scan measurements of the nonlinear refractive index (*n*_2_) for the Ag/Au alloy colloid. The findings revealed that the samples exhibited characteristics of self-defocusing materials, as evidenced by a negative *n*_2_. In the CA behaviors, it was observed that the peak-valley difference ΔT_p−v_ increases with increasing the incident power, as illustrated in Figure 11a.

The CA measurements of the Ag/Au colloidal solution were studied using fs laser pulses. It is noteworthy that the utilization of an HRR laser could potentially induce the creation of a thermal lens within the sample due to thermal effects, resulting in cumulative heating. This heating phenomenon can lead to a temperature distribution within the sample, subsequently altering the spatial distribution of the refractive index. Consequently, such changes could distort the CA Z-scan experimental data, leading to potential inaccuracies in the determination of the *n*_2_.

The separation time between laser pulses of 12.5 ns is less than the thermal characteristic time (t_c_ = $\frac{\omega^{2}}{4D}$), where D is the thermal diffusion coefficient of the sample and ω is the laser beam waist. t_c_ for liquids is ≥40 µs [66]. The number of laser pulses incident on the sample is the important factor affecting the accumulative thermal lens through the scan and can be expressed as follows:(2)$$\frac{1}{f\left(Z\right)}=\frac{{a\;L\;E}_{p}{F_{l}}^{\frac{3}{2}}}{\omega {\left(z\right)}^{2}}\left(1-\frac{1}{\sqrt{N_{p}}}\right)$$
where a is the fitting parameter, a = α (dn/dT)/2κ (π3D)1/2, dn/dT represents the temperature derivative of the refractive index, κ represents the thermal conductivity, E_p_ represents the energy per laser pulse, ω(z) is the radius of the laser beam at the sample, F_l_ is the repetition rate, L is the sample thickness, and N_p_ is the number of laser pulses incident on the sample. Np = t × Fl, where t represents the exposure time for the Ag/Au solution during the scan. In CA Z-scan measurements, the normalized transmittance (ΔTCA) depends on the focal length of the induced lensing, which is given by the following [67,68]:(3)$$\Delta T_{\mathrm{CA}}=1+\frac{2Z}{f\left(Z\right)}$$

Figure 11a and Figure 12a show the experimental results at various excitation wavelengths and incident powers, which are simulated by theoretical fitting, as depicted by the solid lines using Equations (2) and (3). The nonlinear phase shift ∆φ can be obtained by the following [69]:(4)$$\Delta \varphi =\frac{Z_{0}}{2f\left(0\right)}$$
where f(0) represents the focal length of the induced thermal lens when the sample is placed at the focal point (Z = 0). The nonlinear refractive index n_2_ can be deduced by knowing the nonlinear phase, which can be expressed by [69] as follows:(5)$$n_{2}=\frac{\lambda \;{\omega_{0}}^{2}\Delta \varphi }{\left(2P_{p}\times L_{\mathrm{eff}}\right)}$$
where P_p_ is the peak power. The nonlinear refractive index of the colloidal solution as a function of incident power was determined using the best fit of the experimental CA data presented in Figure 11a, using Equations (2)–(5). Figure 11b shows the influence of the incident laser power on the *n*_2_ of the Ag/Au alloy at an 800 nm excitation wavelength. As the incident power increases, so does the number of free carriers in the Ag/Au alloy, resulting in the electronic Kerr effect and thermal accumulative effects, which cause thermal expansion and variations in the polarizability of the Ag/Au alloy.

Figure 12a illustrates the impact of different excitation wavelengths on the *n*_2_ of the Ag/Au alloy NP sample at a 1 W incident power. Higher excitation wavelengths lead to increased peak-to-valley transmission difference (ΔT_p−v_). Subsequently, Figure 12b presents the values of *n*_2_ obtained from Figure 12a plotted against the excitation wavelength for the Ag/Au sample. Notably, Figure 12b demonstrates a linear relationship between the excitation wavelength and the nonlinear refractive index. As the excitation wavelength increases, *n*_2_ also increases, as corroborated by the data presented in Table 1. This behavior can be attributed to the concurrent increase in nonlinear absorption with increasing wavelength, as previously elucidated, given that absorptive processes contribute to nonlinear refraction.

### 3.4. Comparison of the Nonlinear Optical Characteristics of Monometallic (Ag, Au) and Bimetallic (Ag/Au) NPs

Both CA and OA Z-scan measurements were conducted on the Ag, Au, and Ag/Au NPs solutions to calculate the nonlinear refractive index *n*_2_ and nonlinear absorption coefficient γ. In the OA Z-scan, the transmittance decreased (indicating reverse saturable absorption, RSA) as the samples traversed along the beam axis.

Figure 13a portrays the OA Z-scan curves of the Ag, Au, and Ag/Au alloy samples at 800 nm excitation and 1 W incident power. It is evident that the nonlinear absorption coefficient of the Ag/Au alloy sample surpassed that of the Ag and Au NPs samples, as corroborated by Table 1. This variation in the NLO properties of the alloy solution compared to the monometallic Ag and Au NPs can be attributed to bimetallic Ag/Au nanoclusters, which have been demonstrated to possess distinctive characteristics in contrast to their monometallic counterparts.

Figure 13b depicts the CA Z-scan experimental data of the Ag, Au, and Ag/Au alloy NPs colloidal solutions. Notably, the monometallic materials (Ag and Au) exhibited marginally lower nonlinear refraction compared to the Ag/Au alloy sample, as summarized in Table 1.

### 3.5. Optical Limiting Effect of Ag, Au, and Bimetallic Ag/Au NPs

The optical limiting effect of Ag, Au, and Ag/Au alloy NP colloids was systematically investigated by varying the input power at an 800 nm excitation wavelength. Figure 14 illustrates the optical limiting effect observed for the Ag/Au alloy sample across different excitation wavelengths (780, 800, and 820 nm).

During experimentation, the alloy colloid was positioned at the focal point of a 5 cm convex lens, enabling the measurement of output power at various input power levels. The results of optical limiting (OL) analysis revealed a linear relationship between output power and input power until reaching saturation at higher input power levels, a characteristic hallmark of the OL phenomenon.

It is noteworthy that the optical limiting action of the alloy sample exhibited sensitivity to the excitation wavelength, which correlates with nonlinear absorption processes, as discussed previously.

The optical limiting performances of both monometallic and bimetallic nanoparticle samples composed of Au and Ag are illustrated in Figure 15. Notably, the Ag/Au alloy sample demonstrates a notably superior optical limiting effect compared to both Ag and Au individually. This enhancement in optical limiting efficacy is attributed to the higher reverse saturable absorption (RSA) exhibited by the alloy compared to its monometallic counterparts.

Figure 15 also highlights that while the Ag and Au NPs display higher optical limiting saturations compared to the Ag/Au alloy sample, lower values of optical limiting saturation are preferable for effective optical-limiting applications. Consequently, the optical limiting properties of the Ag/Au bimetallic NPs were notably enhanced, characterized by lower optical limiting saturation values relative to both Au and Ag NPs.

Table 2 summarizes the findings from previous studies on the NLO properties of Ag/Au alloy NPs solutions [49,50,51,52,53,70,71,72], alongside the results obtained from the current experimental study. Through our investigation, we deduced that the nonlinear optical behavior and third-order NLO parameters of our Ag/Au NPs samples were significantly impacted by two primary factors: the characteristics of the NPs sample (encompassing nanoparticle shape, size, concentration, and preparation method) and the laser parameters (including excitation wavelength, repetition rate, incident power, and pulse duration). The nonlinear absorption coefficient in this study is three-photon absorption, as in contrast to previous studies that employed two-photon absorption. The aforementioned factors are the causes of the discrepancy.

## 4. Conclusions

In this study, colloidal solutions of Ag, Au, and Ag/Au NPs alloys were created in distilled water, and their linear and nonlinear optical properties were investigated. The synthesis of Ag/Au alloy NPs occurred in two sequential steps. Initially, Ag and Au NPs were colloidalized using a pulsed laser ablation technique at a laser intensity of 71.9 MW/cm^2^ and an exposure time of 30 min. Subsequently, a mixture (1:1) of colloidal Ag/Au NPs underwent irradiation by a pulsed laser. The influence of various laser parameters, encompassing different laser intensities of 6.3, 8.9, and 11.4 MW/cm^2^, as well as different exposure times of 5, 15, and 30 min, on a mixture of colloidal Ag/Au NPs, was meticulously studied.

The formation of Ag/Au alloy NPs was indicated by the emergence of a single peak between Ag and Au NPs at a laser intensity of 11.4 MW/cm^2^ and an exposure duration of 15 min, according to UV–Vis absorption spectroscopy. The combination’s average size dropped to 8.5 nm as the laser intensity reached 11.4 MW/cm^2^, and further diminished to 8 nm with an increase in exposure time to 30 min. Moreover, the average size of a combination of Ag/Au NPs was equivalent to a quantum dot (<10 nm) with heightened laser intensity and exposure time. Furthermore, it was observed that *n*_2_ and *γ* were reliant on wavelength and power. Both monometallic (Ag and Au) and bimetallic (Ag/Au) solutions exhibited reverse saturable absorption (RSA) due to their nonlinear absorption behavior. Additionally, the optical limiting impacts of colloidal solutions of Ag, Au, and Ag/Au were studied, with alloy colloids exhibiting a superior limiting effect compared to monometallic NPs, thus suggesting their potential utility as protectors against undesired laser illumination. These findings underscore the promising applications of these nanostructures in photothermal and nonlinear optics.

## Figures and Tables

**Figure 1 nanomaterials-14-01290-f001:**
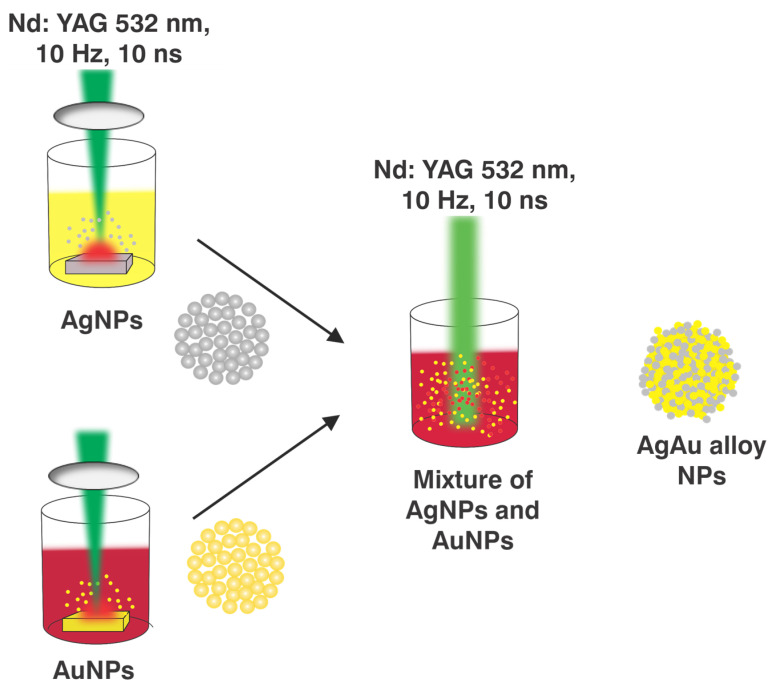
The experimental setup for creating metallic NPs by PLAL.

**Figure 2 nanomaterials-14-01290-f002:**
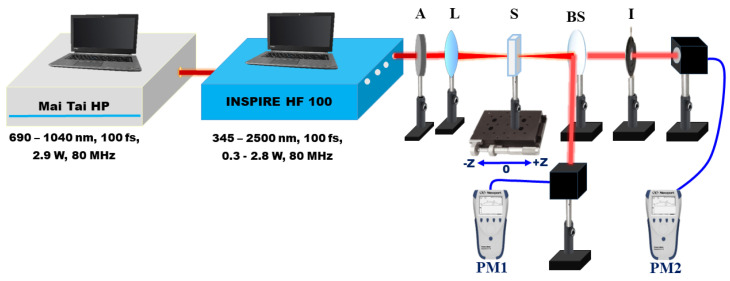
Schematic diagram of the Z-scan experimental setup. A, attenuator; L, convex lens; PM, power meter; S, sample; BS, beam splitter; I, iris.

**Figure 3 nanomaterials-14-01290-f003:**
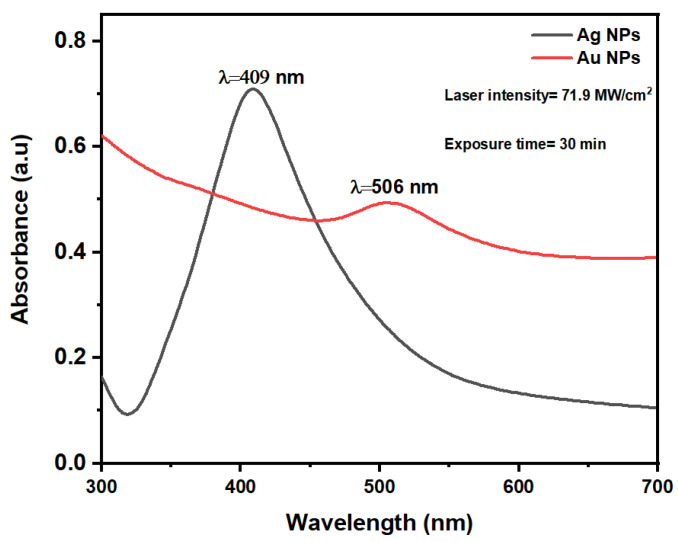
Absorption spectra of Ag and Au NPs after PLAL.

**Figure 4 nanomaterials-14-01290-f004:**
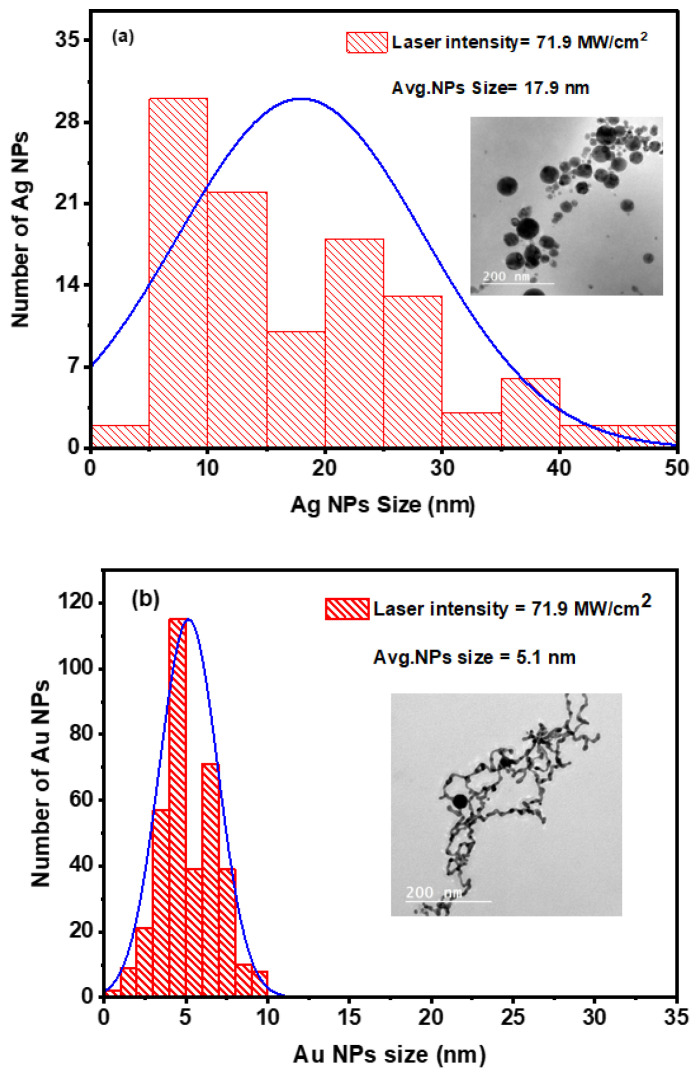
Size distribution histograms for Ag (**a**) and Au (**b**) NPs. The TEM images of the generated Ag and Au spherical NPs are also included in the inset figures.

**Figure 5 nanomaterials-14-01290-f005:**
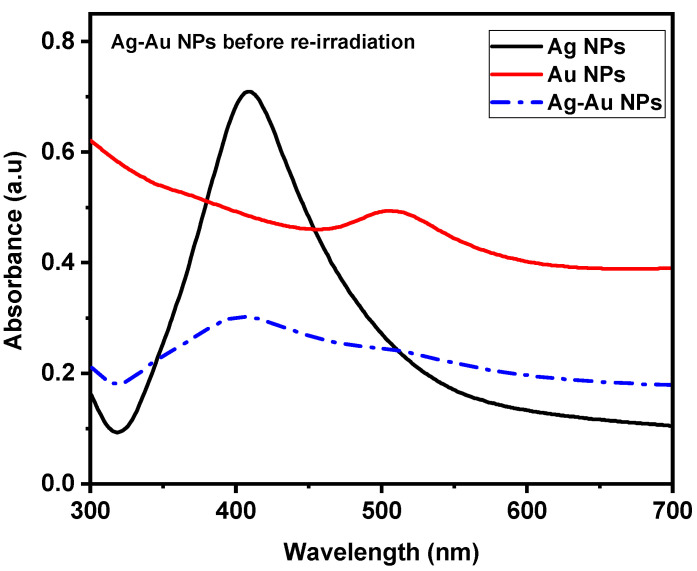
The UV-Vis absorption spectra of Ag and Au NPs prepared by PLAL and a combination of Ag/Au NPs before reirradiation.

**Figure 6 nanomaterials-14-01290-f006:**
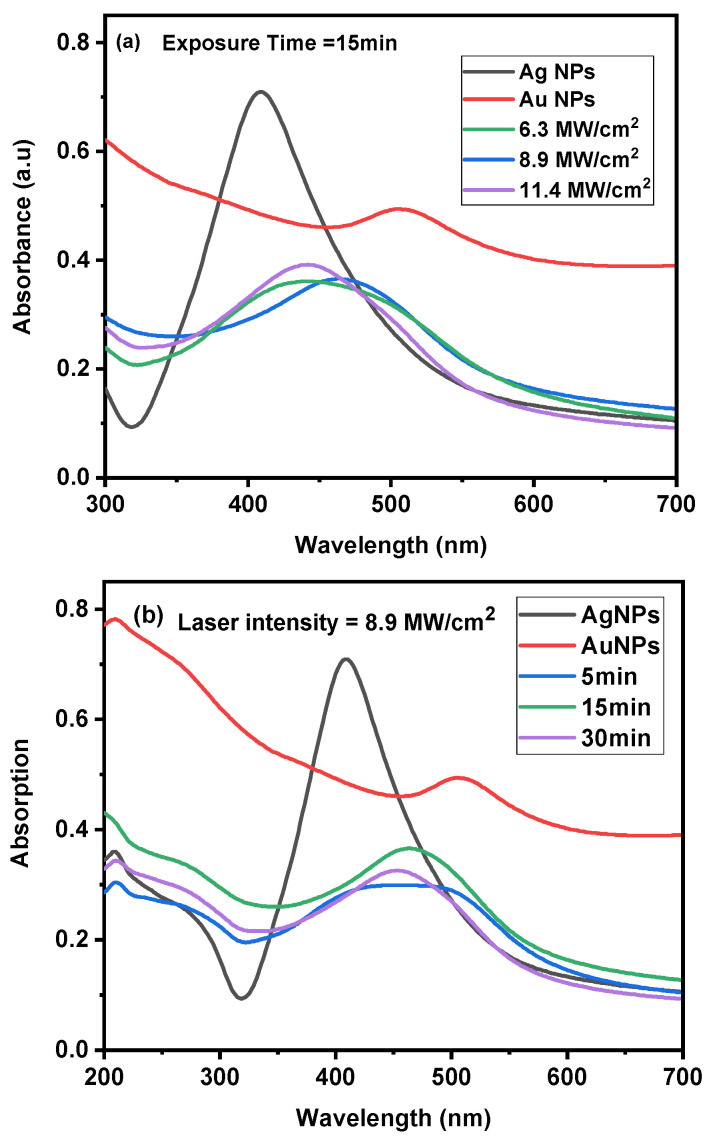
UV-Vis absorption spectra of Ag/Au alloy NPs: (**a**) changing the laser intensity at a constant exposure time of 15 min and (**b**) varying the exposure time at a constant laser intensity of 8.9 MW/cm^2^.

**Figure 7 nanomaterials-14-01290-f007:**
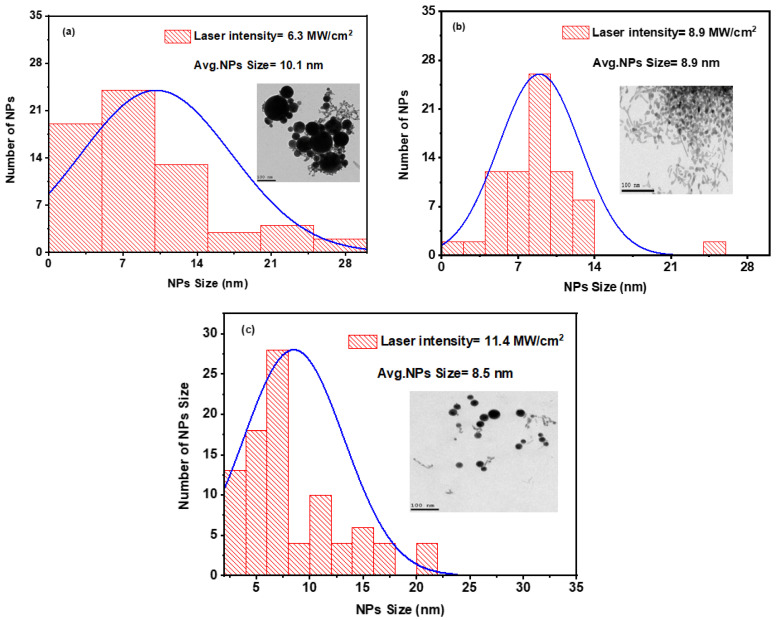
Histogram of the size distribution for a combination of colloidal Ag and Au NPs at different laser intensities: (**a**) 6.3 MW/cm^2^, (**b**) 8.9 MW/cm^2^, and (**c**) 11.4 MW/cm^2^ at an exposure time of 15 min. The insets are TEM images of the colloidal mixture.

**Figure 8 nanomaterials-14-01290-f008:**
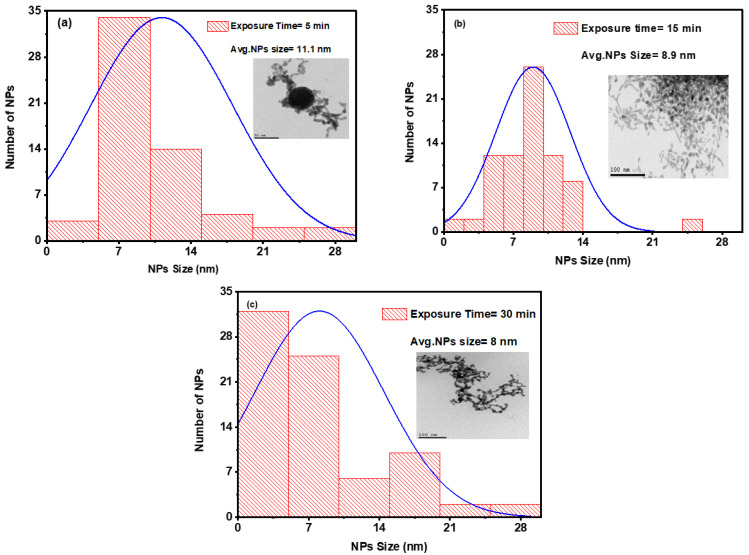
Histogram of the size distribution for a combination of colloidal Ag/Au NPs at three distinct exposure times of (**a**) 5 min, (**b**) 15 min, and (**c**) 15 min at a constant laser intensity of 8.9 MW/cm^2^. The insets are TEM images of a colloidal combination.

**Figure 9 nanomaterials-14-01290-f009:**
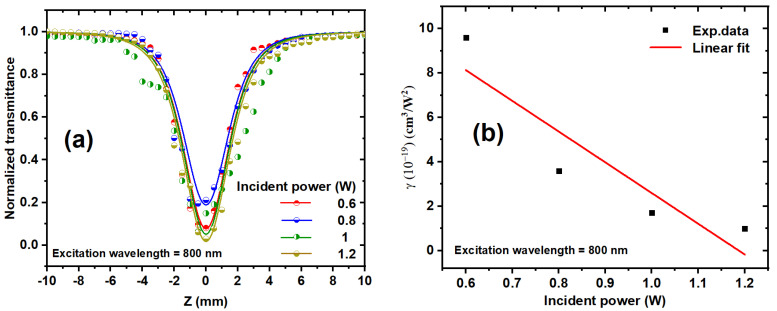
(**a**) OA z-scan transmission for Ag/Au alloy colloids at various incident powers and at an 800 nm excitation wavelength. The dots correspond to the experimental data, and the solid curves represent the fits obtained utilizing Equation (1). (**b**) Relationship between the incident power and the obtained values of γ for the Ag–Au colloidal solution at an 800 nm excitation wavelength. The dots indicate the experimental data, while the solid line represents a linear fit.

**Figure 10 nanomaterials-14-01290-f010:**
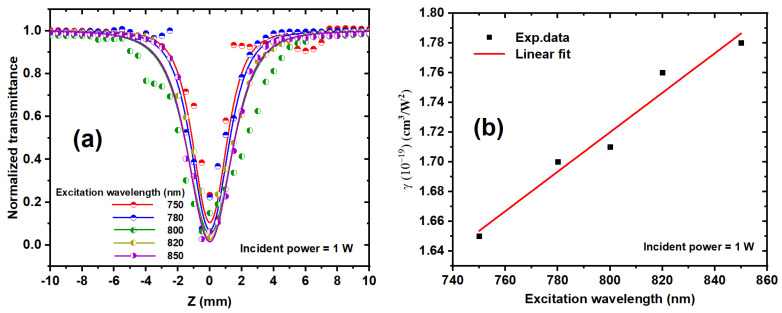
(**a**) Show the OA z-scan transmission for Ag/Au alloy solutions in water at various excitation wavelengths from 750 to 850 and a constant power of 1 W. The solid curves were fit using Equation (1). (**b**) Display the relation between the excitation wavelength and the deduced values of γ for the Ag/Au colloid solution at a constant incident power. The dots represent the experimental data, while the solid line represents the linear fit.

**Figure 11 nanomaterials-14-01290-f011:**
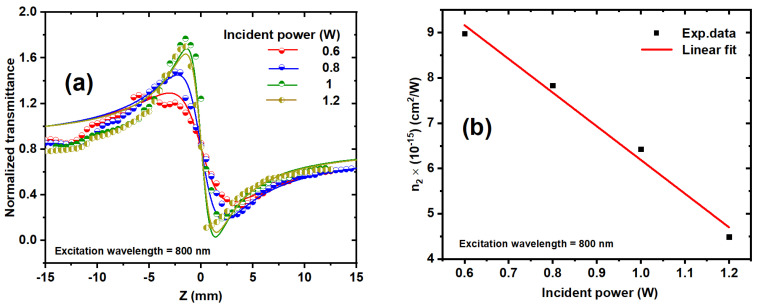
(**a**) The CA z-scan measurements of the Ag/Au NP sample at various incident powers ranged from 0.6 to 1.2 mW at a constant excitation wavelength of 800 nm. The symbols correspond to the experimental data, and the solid curves represent the fits obtained using Equations (2) and (3). (**b**) The deduced *n*_2_ values of the Ag/Au NPs sample as a function of incident laser power at an excitation wavelength of 800 nm. The dots represent the experimental data, and the solid line represents the linear fit.

**Figure 12 nanomaterials-14-01290-f012:**
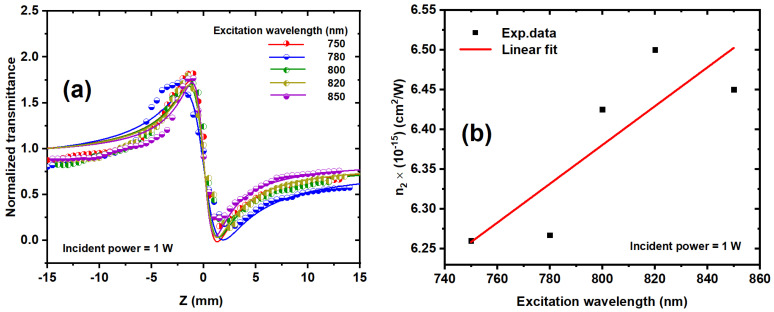
(**a**) CA z-scan measurements for Ag/Au alloy colloids at various excitation wavelengths from 750 to 850 and a 1 W incident power. The solid curves were theoretically fitted using Equations (2) and (3). (**b**) The nonlinear refractive index of the Ag/Au colloid as a function of the excitation wavelength. The dots are the experimental results, and the solid line is the linear fit.

**Figure 13 nanomaterials-14-01290-f013:**
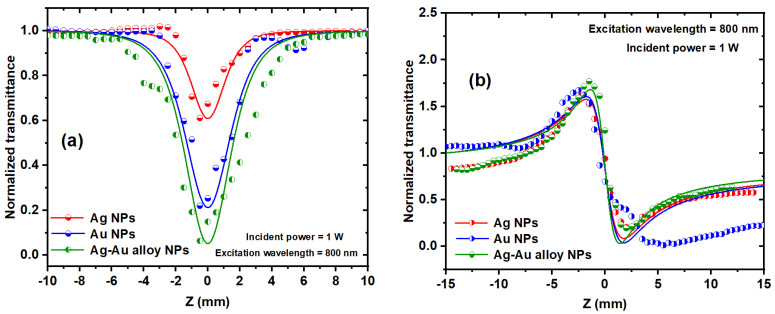
The OA (**a**) and CA (**b**) Z-scan measurements of Ag, Au, and alloy NPs solutions in water at a 1 W incident power and an 800 nm excitation wavelength, respectively.

**Figure 14 nanomaterials-14-01290-f014:**
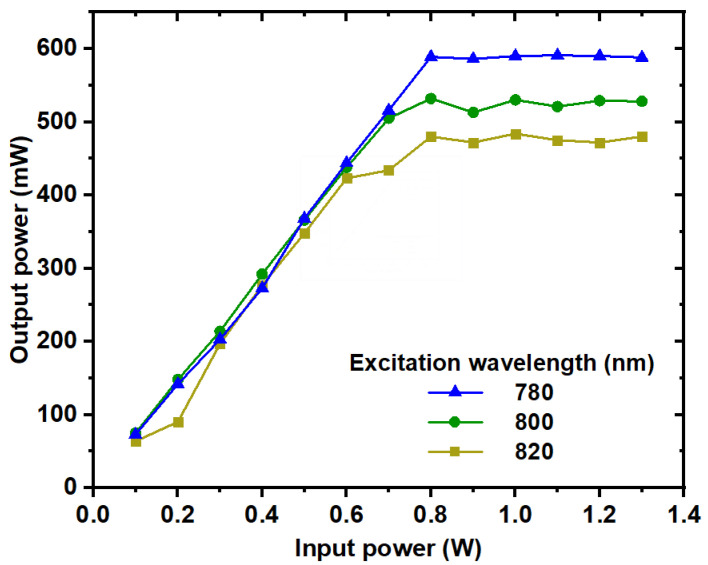
The optical limiting of the Ag/Au alloy colloid at different excitation wavelengths.

**Figure 15 nanomaterials-14-01290-f015:**
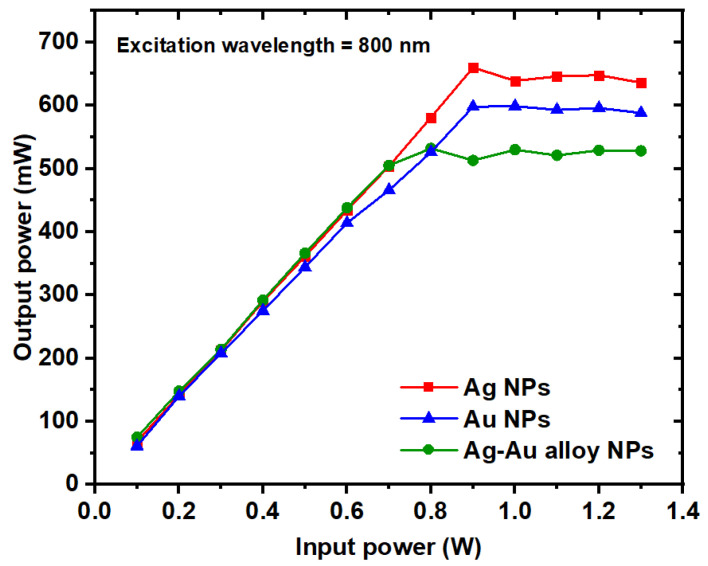
The nonlinear optical limiting performance of the Ag, Au, and Ag/Au alloy samples as a function of the input power at an excitation wavelength of 800 nm.

**Table 1 nanomaterials-14-01290-t001:** Shows the NLO parameters of metal NPs at an 800 nm excitation wavelength and an incident average power of 1 W. The n_2_ represents the nonlinear refractive index of NPs colloids, n_0_ is the linear refractive index, and γ is the 3PA coefficient.

Metal NPs	n_0_	n_2_ × 10^−15^ (cm^2^/W)	γ × 10^−19^ (cm^3^/W^2^)
Ag	2.06	5.82	0.72
Au	4.69	6.19	1.42
Ag–Au	1.96	6.43	1.71

**Table 2 nanomaterials-14-01290-t002:** Comparison between the extracted nonlinear optical properties of Ag/Au bimetallic NPs from this work and previous studies.

Au@Ag Bimetallic NPs	Preparation Method	Wavelength(nm)	Pulse Duration	Repetition Rate (Hz)	Au–Ag Avg. Size (nm)	n_2_ (cm^2^/W)	Ref.
Au–Ag in water	Laser ablation	532	10 ns	10	14.8	−0.5 × 10^−12^	[51]
					16.3	−0.521 × 10^−12^	
					11.6	−0.892 × 10^−12^	
Au@Ag in water	Chemical	632.8	-	-	20	81.6× 10^−6^	[73]
Au@Ag in water	Chemical	1040	357 fs	1000	73	4.41 × 10^−16^	[71]
Au@Ag in water	Laser ablation	800	50 fs	1000	19.8	1.6 × 10^−12^	[72]
Au–Ag in water	Laser ablation	From 750 to 850	100 fs	80 × 10^6^	8.5	From 6.26 × 10^−15^ to 6.45 × 10^−15^	This work

## Data Availability

The data that underlie the results that are presented in this paper are not publicly available at this time but can be obtained from the authors upon reasonable request.

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
