# Peer review of "Using Femtosecond Laser Pulses to Explore the Nonlinear Optical Properties of Ag/Au Alloy Nanoparticles Synthesized by Pulsed Laser Ablation in a Liquid"

_nanomaterials, 2024, doi:10.3390/nano14151290_

Round 1
Reviewer 1 Report
Comments and Suggestions for Authors
In this study, the authors presented the preparation and characterization of colloidal solutions of Ag, Au, and Ag/Au NPs. The synthesis of Ag/Au alloy NPs is based on laser ablation and it occurred in different steps. The authors discussed in details how the parameters used during the synthesis modify the properties of the produced NPs. The optical morphological and optical properties were also discussed in details.
The manuscript is well written and the results convincing. I don't see particular criticisms.
My only comment regards the TEM characterization:
- Fig.4a, Fig8a and 8c, in particular, suggest that , while Ag NPs look quite uniform in size, the Au NPs did not. with some "large size" particles in the area of analysis. these should be better discussed. The average size is not fully convincing.
as a minor note, the authors could consider to mention a recent paper discussing the preparation of AuAg nanostructured plasmonic platform that could be of interest in relation to the topic discussed here (Nanophotonics 2024; 13(7): 1159–1167, 10.1515/nanoph-2023-0942)
Author Response
A point-by-point response to Reviewer #1 comments
Dear,
We appreciate your excellent remarks on our paper, as well as your comments, corrections, and valuable ideas. We believe the following response addresses all of the reviewers' issues. The detailed revisions are listed below where we present the comments of the reviewer in italic red letters, followed by our response in blue letters.
- The manuscript is well written and the results convincing. I don't see particular criticisms.
We would like to thank the reviewer for his kind words and support.
- My only comment regards the TEM characterization:
- Fig.4a, Fig8a and 8c, in particular, suggest that , while Ag NPs look quite uniform in size, the Au NPs did not. with some "large size" particles in the area of analysis. these should be better discussed. The average size is not fully convincing.
We appreciate the reviewer comments, which we have now incorporated into the revised version of the manuscript. The chain-like structure formed in the case of Au NPs is a reported phenomenon [R1, R2]. These nano chains are formed by the coalescence of Au NPs. The physical reason behind the formation of such nano chains is attributed to running the ablation process. In the laser fluences used in this work, nanoparticles of all sizes should be evaporated if they are in the area of laser radiation. This reveals an additional mechanism of supply of an atomic (ions) component that would contribute to the growth of interconnections between nanoparticles and results in the formation of nano chains, as it can be seen from the TEM images.
R1: https://doi.org/10.1016/j.apsusc.2014.02.010
R2: https://doi.org/10.1021/acs.inorgchem.2c01975
- as a minor note, the authors could consider to mention a recent paper discussing the preparation of AuAg nanostructured plasmo utilized [33]. nic platform that could be of interest in relation to the topic discussed here (Nanophotonics 2024; 13(7): 1159–1167, 10.1515/nanoph-2023-0942).
Thank you for the interesting reference. We have added this to the Reference list and believe it is appropriate.

Reviewer 2 Report
Comments and Suggestions for Authors
The paper reviews the fabrication and optical properties of gold, silver, and alloyed particles using laser ablation and investigates the linear and non-linear properties of those particles. The linear properties are mainly represented by measurements of optical absorption over a broad spectral range. The non-linear properties are investigated using the Z-scan technique in the near IR range.
The paper is reasonably well presented and written. However, it will benefit from the following improvements and clarifications.
- Ref 14 is related to semiconducting nanoparticles, or QDs. It is not clear why it needs to be cited here. Semiconductor QDs' properties differ significantly from those of gold and silver.
- Avoid quoting total power, like in line 105 and further through the text. Either quote intensities or power together with the spot size.
- Clarify how the density of the particles in the solution was kept uniform during long measurements. Metal particles tend to percipate at the bottom of the cup, so their density in the solution is not uniform and constant.
- The absorbance spectrum of gold particles doesn't show a well-defined peak. Clarify why this is so and why the peak is well-defined for silver.
- Fig 5. It is not clear why Ag-Au particles would have smaller absorption than the corresponding Au and Ag particles.
- Eq 1. - relates to Tauc's plot. Could you explain why the equation for determining the semiconductor bandgap can be used for metals? Reference 57 is not helpful as it also applies the equation for metals.
- Explain what is meant by band gap here. It is a term used for semiconductors, while it is usually not well-defined for metals. In this study, localised surface plasmon dominates the optical absorption, not the band gap. Unfortunately, Tauc's approach is not good here.
- Please relate your findings of the non-linear optical constants to those of R. Wu in their recent publication on "Localized Plasmon Field Effect of Gold Clusters Embedded in Nanoporous Silicon". Comment on the similarities or discrepancies.
I'll be happy to review this paper once the raised points are addressed.
Comments on the Quality of English LanguageEnglish is readable.
Author Response
A point-by-point response to Reviewer #2 comments
Dear,
We appreciate your excellent remarks on our paper, as well as your comments, corrections, and valuable ideas. We believe the following response addresses all of the reviewers' issues. The detailed revisions are listed below where we present the comments of the reviewer in italic red letters, followed by our response in blue letters.
- The paper is reasonably well presented and written. However, it will benefit from the following improvements and clarifications.
We would like to thank the reviewer for his kind words and support
- Ref 14 is related to semiconducting nanoparticles, or QDs. It is not clear why it needs to be cited here. Semiconductor QDs' properties differ significantly from those of gold and silver.
We appreciate the reviewer comments. R14 was replaced with an appropriate one (Preparation and characterization of Au–Ag and Au–Cu alloy nanoparticles in chloroform. Journal of Materials Chemistry 2003. 13(7): p. 1789-1792)
- Avoid quoting total power, like in line 105 and further through the text. Either quote intensities or power together with the spot size.
We appreciate the reviewer's valuable comment which have been taken into account in the revised version of the manuscript. The average power in nonlinear optical measurements was left unchanged because the intensity changed along the Z-axis throughout open and closed aperture measurements as the spot size changed.
- Clarify how the density of the particles in the solution was kept uniform during long measurements. Metal particles tend to percipate at the bottom of the cup, so their density in the solution is not uniform and constant.
We appreciate the reviewer's valuable comment. The laser-synthesized nanoparticles (NPs) are highly stable, remaining free from aggregation or precipitation for weeks. This well-documented phenomenon is attributed to the excess negative charge on the surface of the NPs, which prevents their aggregation [R3, R4]. As a result, the concentrations of NPs remained consistent across all experiments.
R3: https://doi.org/10.1016/j.jcis.2020.10.025
R4: https://doi.org/10.1016/j.molliq.2023.121712
- The absorbance spectrum of gold particles doesn't show a well-defined peak. Clarify why this is so and why the peak is well-defined for silver.
We appreciate the reviewer's valuable comment which have been taken into account in the revised version of the manuscript. The surface plasmon resonance (SPR) effect is more pronounced in Ag nanoparticles (NPs) compared to Au NPs, resulting in a stronger and well-defined absorption peak [R5]. Additionally, the formation of nano chain structures, as discussed earlier, causes a slight broadening of the SPR peak in Au NPs, which prevents the observation of a sharp peak in their case.
R5: 10.1007/s11468-009-9120-4
- Fig 5. It is not clear why Ag-Au particles would have smaller absorption than the corresponding Au and Ag particles.
We appreciate the reviewer comments. The decrease in the absorbance for the mixed colloids is due to the difference in the volume of NPs used for UV-Vis absorbance spectroscopy. The UV-Vis absorbance spectroscopy of Ag and Au NPs is performed by using 2 ml colloid solution of both NPs. In the case of mixed colloid 1 ml Ag NPs and 1 ml of Au NPs were used to keep the overall volume the same. As the volume of each of the colloid solutions is halved that results in a decrease in absorbance that can be seen in Fig.5.
- Eq 1. - relates to Tauc's plot. Could you explain why the equation for determining the semiconductor bandgap can be used for metals? Reference 57 is not helpful as it also applies the equation for metals.
We appreciate the reviewer comments, which we have now considered in the revised version of the manuscript. In the revised manuscript, we decided to remove Tauc's plots, as well as the associated discussion and references.
- Explain what is meant by band gap here. It is a term used for semiconductors, while it is usually not well-defined for metals. In this study, localised surface plasmon dominates the optical absorption, not the band gap. Unfortunately, Tauc's approach is not good here
We appreciate the reviewer's feedback. Please see point 6's response for context.
- Please relate your findings of the non-linear optical constants to those of R. Wu in their recent publication on "Localized Plasmon Field Effect of Gold Clusters Embedded in Nanoporous Silicon". Comment on the similarities or discrepancies.
We appreciate the reviewer's comments. The current work is completely distinct from the R Wu article, and the similarities are minimal. Briefly, the two articles use distinct materials and preparation procedures. Nonlinear optical properties are measured in different ways; in our study, we used the Z-scan technique to investigate the nonlinear refraction, nonlinear absorption, and nonlinear susceptibility, whereas in the npSi/Au study, they used the ultrafast time-resolved method. The optical limiting impact was not explored in the npSi/Au work, but it was investigated in our study. The nonlinear optical properties of npSi/Au were not affected by the excitation wavelength.

Reviewer 3 Report
Comments and Suggestions for Authors
See the review report attached

Comments on the Quality of English LanguageAuthor Response
A point-by-point response to Reviewer #3 comments
Dear,
We appreciate your excellent remarks on our paper, as well as your comments, corrections, and valuable ideas. We believe the following response addresses all of the reviewers' issues. The detailed revisions are listed below where we present the comments of the reviewer in italic red letters, followed by our response in blue letters.
- The authors present laser production, characterization and a possible field of application of Ag, Au and Ag-Au nanoparticles (NPs) in distilled water. While a great body of data provided, their scientific value is questionable.
We would like to thank the reviewer for his kind words and support
- There is no discussion of measurements accuracies throughout the manuscript. Presence of 3 meaning digits in presented magnitudes of NPs characteristics, e,g, 19.2 mg/L on line 192; 17.9 nm on line 201; 178 nm, 135 nm, 127 nm, 441 nm, etc. on lines from 240 to 262; 2.37, 2.52, and 2.19 on line 284; 10.1 nm on line 307; etc. etc. till the end of the manuscript) must be justified.
We appreciate the reviewer's valuable comment which have been taken into account and highlighted in yellow in the revised version of the manuscript.
- Relevance of the data (spectra) in Fig.6 to their comments on lines 235-264 not obvious.
We appreciate the reviewer's valuable comment which have been taken into account and highlighted in yellow in the revised version of the manuscript.
- The value of conclusions, which follow from data in Fig 9d, goes to zero because the whole range of the band gap variation (3.4 to 3.5 eV) is below a rational measurement accuracy.
We'd like to thank the reviewer for their valuable feedback. We would like to point out that in the revised manuscript, we decided to remove Tauc's plots, as well as the associated discussion and references. Further investigations are necessary, and we will carefully follow the reviewer's recommendation in future works.
- Regarding Eq.[3], i) it is absent in Refs [64,65] and ii) it is incorrect in the case of made measurements because TOA(z=0) < 0.2 (see Figs 10,11). Respectively, the magnitudes of 3Pa coefficient in Figs 10b,11b and in Table 1 are questionable.
We appreciate the reviewer's valuable comment which have been taken into account in the revised version of the manuscript. The references 64 and 65 have been replaced with the appropriate ones. Equation 3 was used to estimate the 3PA using experimental data. The normalized transmittance (TOA) of NPs colloids reflects both nonlinear absorption and nonlinear scattering. The resulting 3PA was found to be comparable to the 3PA results in previous reports [1-3].
- Gu, B., et al., Z-scan theory for material with two-and three-photon absorption. J Optics express, 2005. 13(23): p. 9230-9234.
- Mohamed, T., et al., Excitation wavelength and colloids concentration-dependent nonlinear optical properties of silver nanoparticles synthesized by laser ablation. J Materials, 2022. 15(20): p. 7348.
- Ashour, M., et al., Using femtosecond laser pulses to explore the nonlinear optical properties of Au NP colloids that were synthesized by laser ablation. J Nanomaterials, 2022. 12(17): p. 2980.
- The same conclusions valid regarding CA Z-scan experimental data and n2 following from those, as shown in Figs 12,13 and in Table 1.
We'd like to thank the reviewer for their valuable feedback. Equations 2-5 were used to estimate the n2 using experimental data. The nonlinear optical behavior and third-order NLO parameters of Ag/Au NPs samples were significantly impacted by two primary factors: the characteristics of the Ag/Au NPs samples (encompassing nanoparticle shape, size, concentration, and preparation method) and the laser parameters (including excitation wavelength, repetition rate, incident power, and pulse duration). The nonlinear absorption coefficient in this study is three-photon absorption, in contrast to previous studies that employed two-photon absorption.
Additionally, we would like to emphasize that the presented work in the revised manuscript is preliminary results and that additional investigations are required. We will carefully follow the reviewer's recommendation in future work.
- The magnitude of the radius of the laser beam waist is not provided. This does not allow to verify the made claims regarding the data following from the presented equations.
We appreciate the reviewer's valuable comment which have been taken into account in the revised version of the manuscript. Please refer to line 344 of the revised manuscript, where the laser beam waist is highlighted in yellow.
- There are a lot of mistakes in the list of references. Titles of journals missed in more than half of these, numerous incorrectnesses appear in presentation of authors names abbreviations, Refs [68] and [71] are the same.
We appreciate the reviewer's valuable comments, which have been incorporated into the revised version of the manuscript. All references have been reviewed and corrected.
- Numerous informative and terminological mistakes exist. For example: Table 2 caption claims that the linear refractive index and 3PA coefficient are shown, but these both absent in the Table, the phrase “the photon energy rises with increasing excitation wavelength”, on line 393 is nonconscious, it is not clear what mean words “transmitted intensity of the colloidal solutions” on line 161, ”average and size distributions” on line 204, the spherical structure ..” on line 302, “the average size of the v reduced as the exposure time ..” on line 319. A lot of misprints exist throughout the text.
We appreciate the reviewer's valuable comments, which have been incorporated into the revised version of the manuscript. In the revised manuscript, all the terminological mistakes have been reviewed and corrected.
- Font sizes must be increased in Fig. 7 and in insets in Figs 8&9.
We appreciate the reviewer's valuable comment which have been taken into account in the revised version of the manuscript.

Round 2
Reviewer 3 Report
Comments and Suggestions for Authors
See report attached

Comments on the Quality of English LanguageAuthor Response
A point-by-point response to Reviewer #3 comments
Dear Reviewer,
We appreciate your time, attention, and helpful remarks on the revised manuscript, as well as your suggestions, corrections, and valuable ideas.
We also believe that the changes, made in response to your comments, have improved the quality of the manuscript, and for that we are grateful.
We hope that our response addressed all of your concerns. The detailed changes are listed below, with the reviewer's comments in italic red and our answer in blue letters.
- A discussion and a justification of measurements accuracies is not provided. 3 meaning digits still present in magnitudes of NPs characteristics, radiation intensity, nonlinear refractive index, 3PA coefficient, etc. Error magnitudes added to some magnitudes, but the magnitudes themselves still have 3-4 meaning digits, which looks nonprofessional. For example, 17.9 ± 10 nm on line 209, 218 ± 35 nm on line 266, 135.1 ± 1 nm on line 268, 129.9 ± 3 nm on line 271, etc. etc., looks clumsy.
We appreciate your constructive feedback, which we have attempted to take into account and have indicated in yellow in the revised version of the manuscript.
Regarding your complaints, we believe that in such measurements, the digits are very essential and indicate the precision of the data, which is why we do not want to round the numbers. Furthermore, it makes no sense to change the magnitude of the given number to one or two digits given the unit of measurement. For instance, displaying the laser bandwidth in µm or the nanoparticle particle size in µm will not make sense to readers of the work.
- Regarding Eq.[3], its correctness in the case of made measurements (TOA(z=0) < 0.2 (see Figs 10,11)) is questionable. It is easy to show that it even at T = 0.8 a deviation of γ from a correct magnitude will be of 25% and the deviation increases to ~400% at T = 0.2. Respectively, the magnitudes of 3Pa coefficient in Figs 10b,11b and in Table 1 are questionable. The same conclusions valid regarding CA Z-scan experimental data and n2 following from those, as shown in Figs 12,13 and in Table 1.
We greatly appreciate your informative remark. Regarding your concerns, we noticed a slight discrepancy in fitting the experimental data for both OA and CA. Based on this, Figs 9, 10, 11, 12, and 13 have been revised and reanalyzed carefully in the revised manuscript, as well as the relevant information in Table 1.
Regarding the noted deviation, it is also crucial to emphasize that while fitting the experimental data, we took into account the slight variation in the laser beam radius, which is mainly related to the average power and the excitation wavelength. The tiny variations in the beam radius have a direct impact on the value of the Rayleigh length which consequently affects the value of the 3PA coefficient according to Eq. 1. and the value of n2 as stated in Eq. 2.
- The magnitude of the radius of the laser beam waist provided on line 342 as (19 μm ± 0.2). Regarding this, it is not clear what means “0.2”. If it should be “0.2 μm”, then how such precision is realized? To the best of my knowledge the accuracy of such measurement usually is of 10%.
We appreciate your excellent observation. The magnitude and the unit of the error value have been revised and highlighted in yellow in the revised version of the manuscript.
- Taking this into account a real accuracy of pulse energy and duration measurements, an accuracy of evaluation of radiation intensity cannot be higher than ~15%. As such magnitudes 71.9 MW/cm2 on line 137, 11.4 MW/cm2 online 141, etc. et., do not look reasonable.
We'd like to thank the reviewer for their valuable feedback.
We appreciate your careful remark, but as we go through the peak intensity calculation, it's essential to remember that the beam radius rather than the measurement duration—has a significant impact on the peak intensity value.
To clarify, the laser beam was focused using a convex lens to a beam radius of 1.76mm in this experiment, which resulted in the production of Au and Ag NPs at a beam peak intensity of 71.9 MW/cm2. The convex lens was taken out during the mixture synthesization, exposing the sample to an unfocused beam with a radius of 5 mm. Since the peak intensity in this instance is inversely related to the square of the beam radius, it drops to 11.4 MW/cm2.

Round 3
Reviewer 3 Report
Comments and Suggestions for Authors
The review report attached

Comments on the Quality of English LanguageMinor editing of English language required
Author Response
A point-by-point response to Reviewer #3 comments
Dear Reviewer,
We appreciate your time, attention, and constructive feedback on the revised manuscript, recommendations, corrections, and important ideas.
We also believe that the adjustments made in response to your feedback improved the manuscript's quality, which we appreciate. We hope our response addresses all of your concerns. The detailed revisions are listed below, with the reviewer's criticisms in italic red and our response in blue.
- Regarding Eq. [3], it is an approximate expression, which reasonably well consistent with the classical equation for a material transmittance T altered by the three-photon absorption (3PA), when T > 0.8 only. It is definitely not fulfilled in the case of made measurements (TOA(z=0) < 0.2 (see Figs 10,11)), respectively, the following from Eq. [3] magnitudes of 3PA coefficient in Figs 10b,11b and in Table 1 incorrect too. The same conclusions are valid regarding CA Z-scan experimental data and n2 following from those, as shown in Figs 12,13 and in Table 1. Since, according to the work title, measurement of these characteristics in the goal of study, it made inappropriately
We appreciate your constructive feedback, which we have attempted to consider and have highlighted in yellow in the revised version of the manuscript.
Firstly, we want to point out that Eq. 3 in the original manuscript was changed to Eq. 1 in the revised version of the manuscript. Second, in response to the reviewer's criticism of this particular equation, we replaced it with an additional comprehensive one based on the following references that are listed in the revised version of the manuscript:
- Sheik-Bahae, M., Said, A. A., Wei, T. H., Hagan, D. J., & Van Stryland, E. W. (1990). Sensitive measurement of optical nonlinearities using a single beam. IEEE journal of quantum electronics, 26(4), 760-769.
- Sheik-Bahae, M., Said, A. A., & Van Stryland, E. W. (1989). High-sensitivity, single-beam n2 Optics letters, 14(17), 955-957.
Equation 1, which has been applied in the majority of previous studies, can be applied in the situation when TOA(z=0) < 0.2, as it was in the current work for the OA Z-scan. The following references were used in Equation 1 when TOA(z=0) < 0.2:
- Grehn, M., Seuthe, T., Tsai, W. J., Höfner, M., Achtstein, A. W., Mermillod-Blondin, A., ... & Bonse, J. (2013). Nonlinear absorption and refraction of binary and ternary alkaline and alkaline earth silicate glasses. Optical Materials Express, 3(12), 2132-2140.
- He, J., Qu, Y., Li, H., Mi, J., & Ji, W. (2005). Three-photon absorption in ZnO and ZnS crystals. Optics express, 13(23), 9235-9247.
- Keeping many meaning digits in magnitudes of measured/evaluated characteristics is not an evidence of their accuracy.
We greatly appreciate your informative remark. In response to your comments, we updated the manuscript by changing the number of digits and highlighting them in yellow.
- The magnitude of the radius of the laser beam waist provided on line 343 as (19 μm ± 2), i.e. dimension, “μm”, still missed
We appreciate your profound observation. The magnitude and the unit of the error value have been revised and highlighted in yellow in the revised version of the manuscript.
